# Oleuropein Relieves Pancreatic Ischemia Reperfusion Injury in Rats by Suppressing Inflammation and Oxidative Stress through HMGB1/NF-κB Pathway

**DOI:** 10.3390/ijms251810171

**Published:** 2024-09-22

**Authors:** Maged S. Abdel-Kader, Rehab F. Abdel-Rahman, Gamal A. Soliman, Hanan A. Ogaly, Mohammed A. Alamri, Abdulrahman G. Alharbi

**Affiliations:** 1Department of Pharmacognosy, College of Pharmacy, Prince Sattam Bin Abdulaziz University, Al-Kharj 11942, Saudi Arabia; 2Department of Pharmacognosy, College of Pharmacy, Alexandria University, Alexandria 21215, Egypt; 3Department of Pharmacology, National Research Centre, Giza 12622, Egypt; rf.abdelrahman@nrc.sci.eg; 4Department of Pharmacology and Toxicology, College of Pharmacy, Prince Sattam Bin Abdulaziz University, Al-Kharj 11942, Saudi Arabia; g.soliman@psau.edu.sa (G.A.S.); ma.alamri@psau.edu.sa (M.A.A.); 5Department of Pharmacology, College of Veterinary Medicine, Cairo University, Giza 12613, Egypt; 6Department of Biochemistry, College of Veterinary Medicine, Cairo University, Giza 12613, Egypt; ohanan@kku.edu.sa; 7Maternity and Children’s Hospital, Ministry of Health, Al-Kharj 11942, Saudi Arabia; aalharbi556@moh.gov.sa

**Keywords:** oleuropein, pancreas, ischemia, reperfusion, inflammation, NF-κB, HMGB1, rat

## Abstract

Oleuropein (OLP) is a naturally occurring phenolic compound in olive plant with antioxidant and anti-inflammatory potential and can possibly be used in treating pancreatic injuries. This investigation aimed to follow the molecular mechanism behind the potential therapeutic effect of OLP against pancreatic injury persuaded by ischemia–reperfusion (I/R). Pancreatic I/R injury was induced by splenic artery occlusion for 60 min followed by reperfusion. Oral administration of OLP (10 and 20 mg/kg) for 2 days significantly alleviated I/R-persuaded oxidative damage and inflammatory responses in pancreatic tissue as indicated by the decreased malondialdehyde (MDA) content and increased glutathione peroxidase (GPx) activity, accompanied by the suppression of myeloperoxidase (MPO) activity and reduced levels of interleukin-1beta (IL-1β), nuclear factor kappa B (NF-κB), and tumor necrosis factor alpha (TNF-α) in pancreatic tissues. Furthermore, OLP treatment markedly restored the serum levels of amylase, trypsinogen-activated peptide (TAP), and lipase, with concurrent improvement in pancreatic histopathological alterations. Moreover, treatment with OLP regulated the pancreatic expression of inducible nitric oxide synthase (iNOS) and high-mobility group box 1 (HMGB1) relative to rats of the pancreatic IR group. Thus, OLP treatment significantly alleviates the I/R-induced pancreatic injury by inhibiting oxidative stress and inflammation in rats through downregulation of HMGB1 and its downstream NF-κB signaling pathway.

## 1. Introduction

The susceptibility of the pancreas to ischemia–reperfusion (I/R) injury remains a clinical concern during hemolytic shock, organ surgery, and pancreatic tissue transplantation [1,2]. The pancreas exposure to I/R injury is increased during conditions characterized by decreased abdominal blood flow, such as tissue transplantation, vascular dysfunction, post-traumatic hemorrhage, and thrombosis events [3]. The main cause of pancreatic injury in all these processes is oxidative stress [4]. According to previous studies, reactive oxygen species (ROS) generation can trigger acute pancreatitis (AP), which overwhelms the antioxidant defenses and results in inflammatory cell infiltration, enzymatic increase, and pro-inflammatory cytokine activation [5].

Pancreatic I/R injuries are reported to increase blood and lavage white blood cell count, reactive oxygen species, and inflammatory cytokine release [6]. As a result of acute pancreatitis, the development of systemic inflammatory response syndrome and multiple organ failure, including the pancreas, is a consequence of the release of inflammatory cytokines by activated leukocytes [7]. Moreover, acute pancreatitis can result in the abnormal activation of pancreatic enzymes, which can induce the proteolytic activation of apoptotic factors persuading cell injury or death in lung tissue [8].

Several experimental and clinical investigations showed that the pancreas is extremely vulnerable to ischemic damage. Studies also indicated that modifications in microvascular perfusion occur shortly following the induction of acute experimental pancreatitis [9].

Oleuropein (OLP) is a bioactive phytoconstituent in olive-derived materials and has been reported as the main polyphenol found in olive leaves and oil [10]. OLP expresses antioxidant properties via scavenging of the free radicals. OLP possesses various pharmacological actions, including antibacterial, antiviral, and antifungal [11]. Previous research revealed that OLP is a potent anticancer, hypotensive, and hypoglycemic agent and protects against heart diseases [12]. OLP suppressed cisplatin-induced degenerative changes in the pancreas by inhibiting lipid peroxidation and reactive oxygen species generation [13].

Thus, the present investigation was designed to explore the possible protective effects of OLP during acute pancreatitis caused by pancreatic ischemia/reperfusion in rats.

## 2. Results

### 2.1. Effect on Pancreatic Digestive Enzymes Activities in Serum

The serum levels of amylase, lipase, and trypsinogen-activated peptide (TAP) are recorded in Table 1. Compared to the sham group, amylase, lipase, and TAP significantly increased in the IR control group. Treatment with OLP (Figure 1) causes those enzymes to revert closer to their basal levels. This protective effect was more pronounced after a 20 mg/kg OLP dose.

### 2.2. Effect on Oxidative Stress Markers in Pancreatic Tissues

MDA and MPO concentrations were observed to be higher in pancreatic tissues of the IR control group in comparison to the sham group (1.9 ± 0.04 vs. 0.3 ± 0.02 nmol/mg protein and 7.5 ± 0.31 vs. 0.9 ± 0.06 ng/mg protein, respectively). Further, a significant decrease in GPx activity was detected in the pancreatic tissue of IR control rats compared with the sham group (Table 2). The elevated values of MDA and MPO in the pancreatic tissue were restored and GPx activity was significantly improved in all OLP-treated groups compared to IR control rats.

### 2.3. Effect on Inflammatory Markers in Pancreatic Tissue

The results of Table 3 show a significant increase in levels of pancreatic inflammatory markers, TNF-*α*, IL-1*β*, and NF*κB*, in IR control rats compared to the sham group. On the other hand, OLP-treated rats showed decreased levels of TNF-*α*, IL-1*β*, and NF*κB* compared to IR control rats.

### 2.4. HMGB1 Gene Expression Analysis

The current findings reveal that the I/R control rats showed an upregulated HMGB1 expression in the pancreas as evidenced by the significant (*p* ≤ 0.05) higher fold change in the HMGB1 mRNA level in this group relative to the sham group (Figure 2). Interestingly, the pancreatic HMGB1 expression significantly decreased after receiving OLP-10 and OLP-20 treatments compared to I/R control rats.

### 2.5. Histopathological Examination of Liver Tissues

As depicted in Figure 3, the hepatic tissue of the sham group showed a normal histological structure, while hepatic sections of IR controls revealed congestion of portal blood vessels and infiltration of the portal area by a low number of mononuclear inflammatory cells with the presence of vacuolar degeneration in some hepatocytes. OLP-treated groups showed activation of Van Kupffer cells and the presence of some mononuclear inflammatory cells.

### 2.6. Histopathological Examination of Pancreatic Tissue

The histopathologic study of the pancreas in the IR control group showed hemorrhage and edema between pancreatic acini (Figure 4). Marked necrobiotic changes were also observed in some pancreatic acini. These findings were considered as evidence of an established AP. OLP-treated groups showed congestion of pancreatic blood vessels with pancreatic edema.

### 2.7. Immunohistochemical Examination of iNOS in Pancreatic Tissue

Photographs of the immunohistochemical investigations of iNOS in pancreatic tissue sections and iNOS expression percent are displayed in Figure 5 and Figure 6. The sham group showed a negative reaction for iNOS. Sections of the pancreatic IR control group showed a strong positive reaction for iNOS in nuclei of pancreatic acini. Pancreatic tissue sections of OLP-treated groups showed a mild to moderate positive reaction for iNOS in the cytoplasm of pancreatic acini.

## 3. Discussion

Pancreatic tissue is highly susceptible to ischemic damage [14]. In the present study, pancreatic I/R was experimentally persuaded in rats by transient occlusion of the splenic artery with a microvascular clamp for 60 min, which was then removed to allow for pancreatic reperfusion. The development of pancreatic I/R has been associated with the activation of the enzymes amylase, lipase, and trypsinogen-activated peptide (TAP) in the blood. The pancreas typically releases these enzymes into the duodenum. In acute pancreatitis induced as a result of pancreatic I/R, these enzymes are released into the pancreatic interstitial space and then into blood circulation [15]. For this reason, enzymes derived from pancreatic acinar cells (amylase and lipase) are the hallmarks in the laboratory diagnosis of acute pancreatitis [15]. Trypsinogen-activated peptide (TAP) is currently under evaluation as an additional biomarker for the diagnosis of acute pancreatitis.

In this study, rats of the IR control group showed high serum levels of amylase, lipase, and TAP compared with sham group rats. Treatment with OLP normalized serum pancreatic digestive enzyme activity in rats with pancreatic I/R. Previous studies have shown that OLP modulates the activity of several enzymes involved in digestion in vitro, which is consistent with our findings. For instance, OLP stimulates pepsin’s enzymatic activity in vitro and hence works as an activator of protein digestion [16].

Oxygen free radicals, disturbance of intracellular homeostasis, and failure of microvascular perfusion appear to be important pathological mechanisms involved in ischemia/reperfusion-induced acute pancreatitis [14]. Numerous experimental investigations have demonstrated that oxidative stress is the fundamental mechanism of I/R injury in the pancreas [3]. Furthermore, in multiple clinical studies, disease severity is dependent on the increase in oxidative stress in patients with acute pancreatitis [17,18].

In this study, pancreatic I/R persuaded intense oxidative stress, characterized by increased free-radical-procured products (MDA) and the depletion of antioxidant enzymes (GPx) in addition to a significant elevation in myeloperoxidase (MPO) enzyme activity in the pancreas. These parameters correlate well with the disease severity in both clinical and experimental studies [17,19]. In this respect, animal models of acute pancreatitis have shown a marked increase in lipid peroxidation products in the pancreas and plasma together with a decrease in the levels of reduced glutathione in the tissue. This suggests increased oxidative stress both at a tissue and systemic level during acute pancreatitis [20]. Lipid peroxidation caused by ROS directly deteriorates cell membranes. Consequently, the formation of toxic products like MDA might worsen cellular damage or function as a chemoattractant to initiate the systemic inflammatory response syndrome. This suggests that MDA may function as a signal for severe pancreatitis [21]. Thus, an increase in levels of MDA is directly linked with tissue injury and organ dysfunction in acute pancreatitis.

An ability of OLP is to act as a free radical scavenger responsible for the protective antioxidant property [22]. In this study, OLP showed a significant enhancement in the oxidative status of pancreatic tissue via reducing free-radical-derived products (MDA) and restoring the antioxidant status of the pancreas exposed to IR in comparison to the IR control group. OLP-20 restored GPx activities in the pancreas to levels similar to those in the sham group. These findings are corroborated by the fact that OLP’s antioxidant potential is primarily related to its capacity to increase radical stability by creating an intramolecular hydrogen bond between the free hydrogen of the hydroxyl group and its phenoxyl radicals [23]. The potent antioxidant activity of OLP is mainly attributed to the hydroxyl groups within its chemical structure [24]. Additionally, there is evidence that OLP stimulates the expression of intracellular antioxidant enzymes. This evidence shows that OLP reduces the production of intracellular ROS and increases the expression of heme oxygenase-1 (HO-1) by triggering the transcription of the NF-E2-related factor 2 (Nrf2) gene [25]. Further studies in alloxan-diabetic rabbits revealed that OLP increased the levels and activity of non-enzymatic antioxidants such as α-tocopherol, glutathione, β-carotene, and ascorbic acid, as well as enzymatic antioxidants like superoxide dismutase (SOD), glutathione peroxidase, and glutathione reductase [26]. Polyphenols, flavonoids, and other naturally occurring substances all have inhibitory effects on MPO [27].

In essence, inflammation is a defensive response persuaded by tissue injury. The injury caused by ischemia–reperfusion triggers the initiation of inflammatory pathways and the activation of vasoactive substances [28]. The injured tissues release inflammatory cytokines or mediators such as interleukins (IL)-1, IL-6, IL-8, IL-17, tumor necrosis factor alpha (TNF-α), monocyte chemoattractant proteins (MCP-1s), cyclooxygenase (COX), iNOS, metalloproteinases (MMPs), and adhesion molecules. IL-6 was found to have the best sensitivity and specificity for the early assessment of severe acute pancreatitis [29]. Various types of cells, including monocytes, macrophages, endothelium, and fibroblasts, produce IL-6 in response to potent pro-inflammatory stimuli like TNF-α and IL-1*β* [30]. The role of IL-6 in the early and accurate prediction of the severity of acute pancreatitis has been reported in multiple studies [16,30]. Furthermore, nuclear factor kappa B (NF-κB) occupies a vital upstream position where it affects the production of numerous pro-inflammatory mediators [31,32]. Consequently, targeting and inhibiting the activities of NF-κB and its downstream inflammatory mediators has the potential to be a promising approach in the development of new anti-inflammatory agents.

In this context, the anti-inflammatory effect of OLP was demonstrated to be at least partially related to its antioxidant properties [33]. This anti-inflammatory effect was found to be mediated via inhibitory action on NF-κB translocation to the nucleus, cyclo-oxygenase-2 (COX-2), caspase-3, and iNOS [34]. OLP was also found to attenuate the release of several inflammatory cytokines, such as NF-κB, TNF-α, IL-8, and prostaglandin E2, in addition to restoring the expression of protein kinase A (PKA) in the lung of carrageenan-treated mice, and also in a spinal-cord-injured mouse model [35]. In the current study, the induction of pancreatic IR injury resulted in a significant elevation in pancreatic inflammatory cytokines, TNF-α, IL1β, and NF-κB, in IR control rats compared to the sham group.

In contrary, OLP-treated rats revealed a significant suppression of inflammatory biomarkers in pancreatic tissue compared to the IR control group. OLP was demonstrated to exhibit an anti-inflammatory effect that may be partially attributed to its antioxidant capabilities [33]. According to Domitrović et al. (2012) [34], the molecular perspective shows that the anti-inflammatory impact of OLP is mediated through the suppression of NF-κB, cyclooxygenase-2 (COX-2), caspase-3, and iNOS.

HMGB1 is a group of nuclear binding proteins providing multiple nuclear, extracellular, and pro-inflammatory functions depending on their subcellular location [36]. Extracellular HMGB1 is either passively released or actively secreted in response to necrotic or inflammatory signals, respectively. HMGB1 acts as a pro-inflammatory reactive factor to trigger inflammation and immune responses [37,38]. Redox-induced HMGB1 activates TLRs and RAGE, which, in turn, lead to the activation of a sustained pro-inflammatory pathway, including the NF-κB pathway, with the subsequent induction of inflammatory genes, including HMGB1 [39]. In the present study, the pancreatic HMGB1 mRNA level showed a significant elevation during IR injury, which suggests the induction of oxidative stress, inflammation, and apoptosis [40]. At the same time, OLP treatment downregulated the expression of HMGB1, suggesting the anti-inflammatory and protective potential of OLP in the postischemic injury. Histopathological examination of pancreatic tissue sections of the IR control group revealed hemorrhage between pancreatic acini with edema, with necrobiotic changes in some pancreatic acini. In line with our findings, Fujimoto et al. (1997) [41] stated that interstitial edema and interstitial cell infiltration are the hallmarks of acute edematous pancreatitis. Acinar cell apoptosis may be triggered by ischemia/reperfusion, and ischemia/reperfusion damage on the pancreas in rats had several hallmarks of acute pancreatitis. Apoptosis in pancreatic acinar cells may be one of the distinctive characteristics of the ischemia/reperfusion damage. However, OLP-treated groups showed mild congestion of pancreatic blood vessels with pancreatic edema.

Furthermore, in the rat model of severe acute pancreatitis, the spleen aggravates numerous organ damage and systemic inflammatory responses. In addition, rats with severe acute pancreatitis showed a significant increase in the ultrastructural damage to the hepatic tissue, as evidenced by the following indicators: necrosis of the cells, lysosome accumulation in Kuppfer cells, edema of the hepatocytes, mitochondrial swelling, nucleolar pyknosis, apoptosis of the cells, and cell necrosis at 15 h as determined by transmission electron microscopy [42]. These results corroborate our observations that the IR control revealed portal blood vessel congestion, as well as the presence of vacuolar degeneration in hepatocytes and low numbers of mononuclear inflammatory cells infiltrating the portal area.

The activation of iNOS and the cytokine cascade triggered by pancreatic ischemia/reperfusion has been reported to cause a systemic inflammatory response and cause severe acute necrotizing pancreatitis with a high death rate. This suggests that NO overproduction by iNOS is correlated with the apoptotic process in the lung and pancreas [43].

The pancreatic IR group showed a strong positive reaction for iNOS in the nuclei of pancreatic acini. Pancreatic tissue sections of OLP-treated groups showed a mild to moderate positive reaction for iNOS in the cytoplasm of pancreatic acini. Previous studies have shown that OLP inhibits the expression of inducible nitric oxide synthase (iNOS) [31,44], especially under anoxia stress [45].

## 4. Materials and Methods

### 4.1. Chemicals

Oleuropein (OLP) (powder form, 98% by HPLC) was procured from Aktin Chemicals, Inc. (Chengdu, China). Ketamine (Sigma Tec., Giza, Egypt), tween 80 (PioChem, Giza, Egypt), 3,3diaminobenzidine (DAB) from Sigma-Aldrich, MA, USA, hematoxylin and eosin stain (H&E) was purchased from Merck (Darmstadt, Germany).

### 4.2. Animals

Male Wistar rats (180–200 g) were obtained from the Animal Unite at the National Research Centre, Egypt. Animals were kept in standard cages under pathogen-free conditions and maintained at a controlled temperature and in normal light–dark cycles. Standard food and water ad libitum were provided. Rats were kept for one week to adapt to the conditions before beginning the experimental protocol. Experiments were performed according to the National Regulations of Animal Welfare and the Institutional Animal Ethical Committee (IAEC), Approval no. 2416072022.

### 4.3. Experimental Design

Twenty-four male rats were distributed into 4 groups.

Sham group: Rats received 1 mL of the vehicle (2% Tween 80 in sterile saline).IR control group: Rats received orally 1 mL of the vehicle kg (24 and 48 h prior to pancreatic IR, and 24 h post pancreatic reperfusion).OLP-10 group: Rats were treated orally with OLP at 10 mg/kg (24 and 48 h prior to pancreatic IR, and 24 h post pancreatic reperfusion).OLP-20 group: Rats were treated orally with OLP at 20 mg/kg (24 and 48 h prior to pancreatic IR, and 24 h post pancreatic reperfusion).

### 4.4. Induction of Pancreatic I/R

One hour after the second dose of the vehicle or OLP, rats were anesthetized with ketamine (50 mg/kg i.p.). The splenic artery was exposed by a midline abdominal incision. The splenic artery obstruction (SAO) was performed with a microvascular clamp to induce pancreatic ischemia. Ischemia was confirmed by the blanching of the pancreatic tissue [46]. After 60 min, the clamp was removed and pancreatic reperfusion was observed visually. The abdominal incision was sutured with silk suture and the rats were allowed to recover [7]. Rats of the sham group were subjected to the same surgical method, with the exception of SAO.

One hour after the additional dose of the vehicle or OLP, blood samples were collected from the retro-orbital venous plexus to estimate serum biochemical parameters. The serum was separated by centrifugation at 1538× *g* for 10 min. Then, rats were euthanized by decapitation and their pancreases and livers were separated. Each pancreas was divided into two portions. The first portion was immediately snap-frozen in liquid nitrogen and stored at −80 °C for gene expression analysis while the second portion was immediately homogenized in ice-cold 10% (*w*/*v*) phosphate buffer. The homogenate was centrifuged at 1800× *g* for 10 min at 4 °C. The supernatant was used for different biochemical analyses.

### 4.5. Serum Analyses

Serum amylase, lipase, and trypsinogen-activated peptide (TAP) were estimated according to the manufacturer’s protocol of the of the used kit (LSBio Incorporated, Shirley, MA, USA).

### 4.6. Assessment of Oxidative Stress and Inflammatory Markers in Pancreatic Tissues

Different oxidative stress and inflammatory markers, such as malondialdehyde (MDA), glutathione peroxidase (GPx), myeloperoxidase (MPO), nuclear factor kappa B (NF-κB), tumor necrosis factor alpha (TNF-α), and interleukin-1beta (IL-1β), were assessed in pancreatic homogenates using ELISA technique according to the manufacturer’s protocol (BioVision Incorporated, Milpitas, CA, USA).

### 4.7. RNA Purification, cDNA Synthesis and Quantitative Real-Time PCR

The effect of OLP on the pancreatic I/R injury was assessed by studying the mRNA expression profile of HMGB1. Total RNA was extracted from the pancreatic tissues of rats in each group using TRIzol reagent (Invitrogen, Waltham, MA, USA). Isolated RNA was reverse-transcribed into cDNA using Revert Aid first strand cDNA synthesis kit (Thermo Fisher Scientific, Inc., Vilnius, Lithuania) according to the manufacturer’s instructions. Quantitative real-time PCR (qRT-PCR) assay was employed in the Maxima SYBR Green/ROX qPCR kit (Thermo Scientific, Waltham, MA, USA) according to the manufacturer’s protocol, with the specific primer sets selected based on NCBI mRNA database for HMGB1 (accession number: NM_001409387.1) and glyceraldehyde 3-phosphate dehydrogenase; GAPDH (accession number: NM_001394060.2). PCR amplification was validated by analyzing the melting curves. The mRNA expression level was determined as relative fold change from untreated control group using the cycle threshold (CT) formula 2^−△△CT^, where GAPDH was used as reference gene for normalization.

### 4.8. Histopathological Examination of Pancreatic and Liver Tissues

Pancreatic and liver tissues were fixed in 10% neutral-buffered formalin for 24 h, washed with tap water, and prepared and stained for light microscopy. For dehydration, serial dilutions of alcohol were applied and, thereafter, the specimens were cleared in xylene and embedded in paraffin wax in hot air oven at 56 °C for 6 h. Paraffin wax tissue blocks were sectioned by using microtome at 5–6-micron thickness. Then, sections were collected on glass slides and deparaffinized. They were stained for routine histological examination using hematoxylin and eosin stain [47].

### 4.9. Immunohistochemical Examination of iNOS in Pancreatic Tissues

Paraffin sections were mounted on positively charged slides by using avidin biotin- peroxidase complex (ABC) method according to Saeedan et al. (2023) [48]. Rabbit inducible nitric oxide synthase (iNOS) polyclonal antibody (Gene tex, Cat.No. GTX130246, Dil.: 1:100) was used. Sections from each group were incubated with these antibodies, and then the reagents required for ABC method were added (Vectastain ABC-HRP kit, Vector Laboratories, Newark, CA, USA). Marker expression was labeled with peroxidase and colored with diaminobenzidine (DAB, produced by Sigma-Aldrich, Burlington, MA, USA) to detect antigen–antibody complex. Negative controls were included using non-immune serum in place of the primary or secondary antibodies. Immunohistochemical-stained sections were examined via using Olympus microscope (BX-53) (Tokyo, Japan). Scoring was performed by determination of reaction area percentage using ImageJ 1.53K, National Institute of Health, Bethesda, ML, USA.

### 4.10. Statistical Analysis

All data were presented as mean ± SEM. Statistical analysis was conducted using one-way analysis of variance (one-way ANOVA) followed by Tukey’s test to determine the intergroup variability by using GraphPad Prism^®^ software (version 6.00 for Windows, San Diego, CA, USA). A probability level of less than 0.05 was accepted as statistically significant.

## 5. Conclusions

To the best of the authors’ knowledge, this is the first investigation on the protective effect of OLP on experimental pancreatic I/R injury. Collectively, our data indicate that OLP significantly alleviated pancreatic injury in a pancreatic I/R rodent model by effectively inhibiting oxidative stress and inflammation. Importantly, the current study provides new insight into the mechanism underlying the pancreatic protection by OLP. Downregulation of HMGB1 and its downstream NF-κB signaling pathway by OLP may partially mediate its anti-inflammatory activity and inhibit the production of pro-inflammatory mediators. We suggest that the development of novel therapies using OLP could be promising for future medical interventions of pancreatic I/R damage.

## Figures and Tables

**Figure 1 ijms-25-10171-f001:**
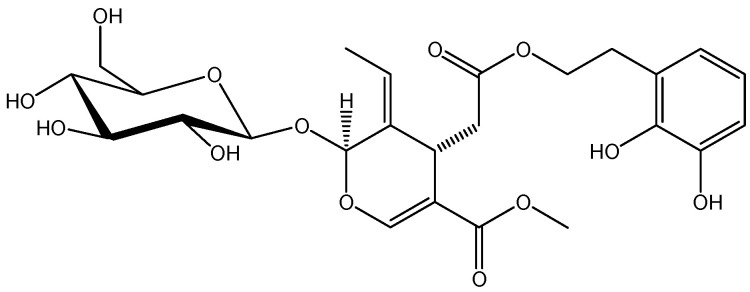
Structure formula of oleuropein (OLP).

**Figure 2 ijms-25-10171-f002:**
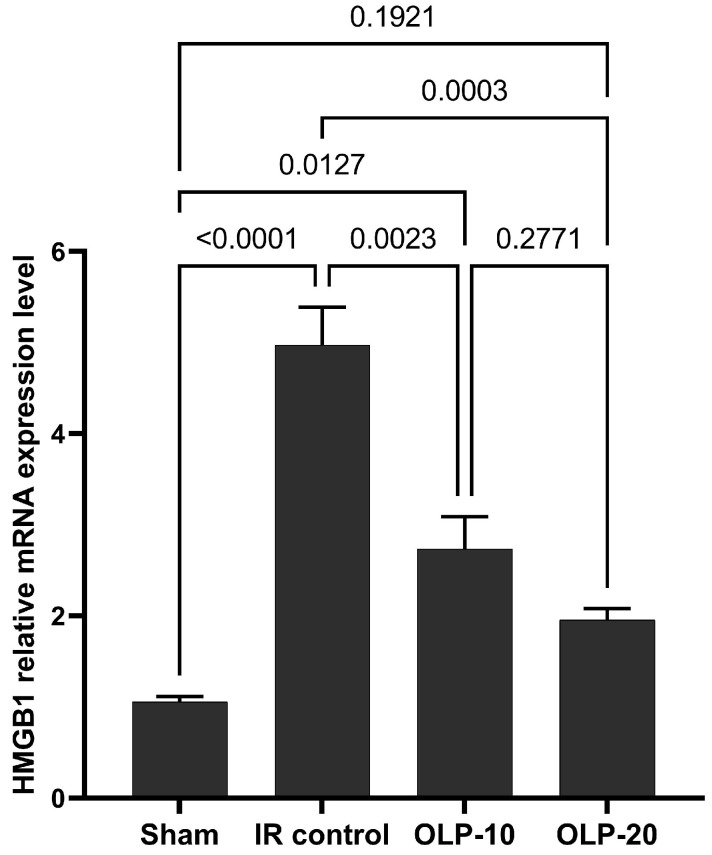
Effect of OLP on HMGB1 gene expression.

**Figure 3 ijms-25-10171-f003:**
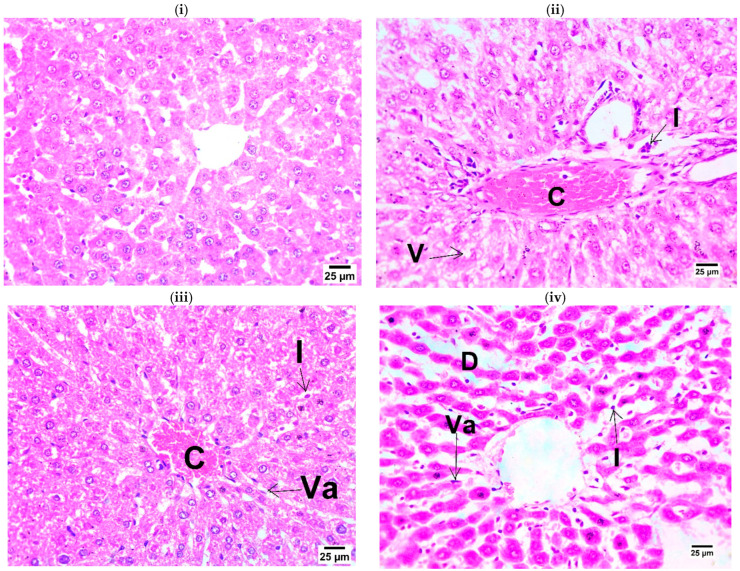
Photomicrographs of rats’ liver (stained with H&E × 400): (**i**) **Sham** showing normal histological structure of liver; (**ii**) **IR control** showing congestion of portal blood vessels (C) with infiltration of portal area by low number of mononuclear inflammatory cells (I) and presence of vacuolar degeneration in some hepatocytes (V); (**iii**) **OLP-10** showing congestion of central vein (C) with activation of van Kupffer cells (Va) and presence of low number of mononuclear inflammatory cells (I) in hepatic sinusoids; and (**iv**) **OLP-20:** photomicrograph showing dilatation of hepatic sinusoids (D) with activation of Van Kupffer cells (Va) and presence of low numbers of mononuclear inflammatory cells (I) in sinusoids.

**Figure 4 ijms-25-10171-f004:**
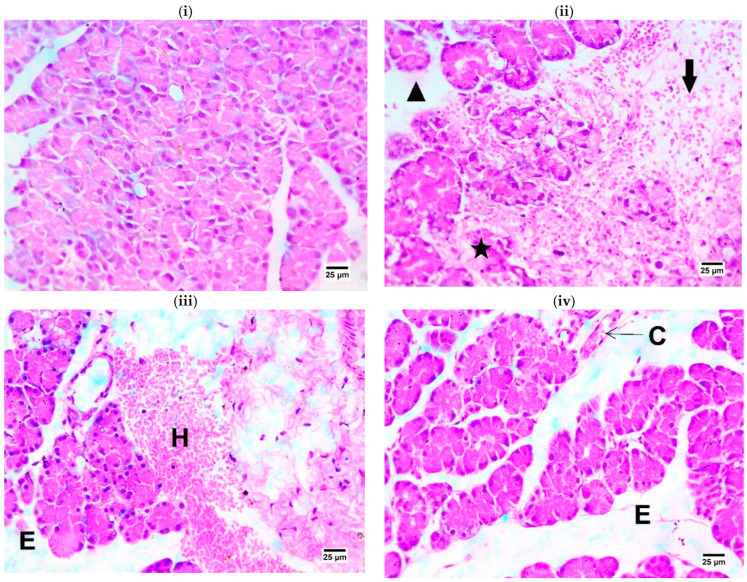
Photomicrographs of rats’ pancreas (stained with H&E × 400): (**i**) **Sham** showing normal histological structure of pancreas; (**ii**) **IR control** showing hemorrhage between pancreatic acini (arrow) with edema (arrow head), necrobiotic changes in some pancreatic acini (star). Presence of necrobiotic changes in some pancreatic acini; (**iii**) **OLP-10** showing hemorrhage between pancreatic acini (H) with edema (E); and (**iv**) **OLP-20:** photomicrograph showing congestion of pancreatic blood vessels (C) with pancreatic edema (E).

**Figure 5 ijms-25-10171-f005:**
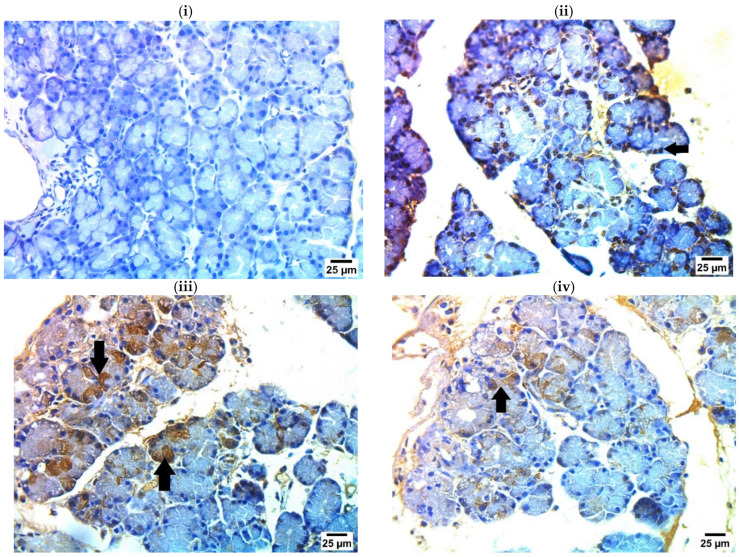
Immunohistochemical examination of iNOS in pancreatic tissue (stained with IHC-peroxidase–DAB × 400): (**i**) **Sham** showing negative reaction for iNOS in pancreatic acini; (**ii**) **IR control** showing strong positive reaction for iNOS in nuclei of pancreatic acini (arrow); (**iii**) **OLP-10** showing moderate positive reaction for iNOS in cytoplasm of pancreatic acini (arrows); and (**iv**) **OLP-20** showing very mild positive reaction for iNOS in cytoplasm of pancreatic acini (arrow).

**Figure 6 ijms-25-10171-f006:**
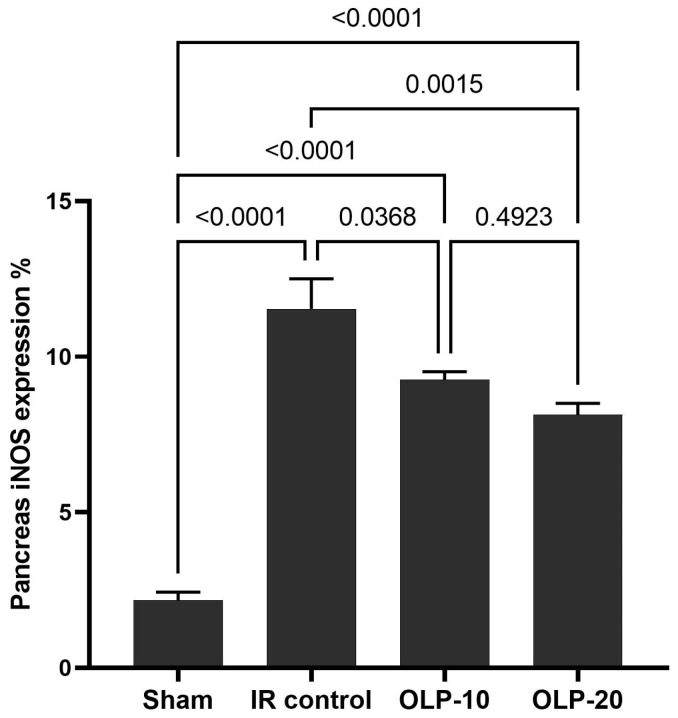
Effect of OLP on pancreatic iNOS.

**Table 1 ijms-25-10171-t001:** Effect of OLP on digestive enzyme activities in serum of rats subjected to pancreatic ischemia/reperfusion (I/R).

Group	Amylase (ng/mL)	*p* Value	Lipase (pg/mL)	*p* Value	TAP (ng/mL)	*p* Value
**Sham**	4.8 ^b^ ± 0.27	<0.0001	27.8 ^b^ ± 1.31	<0.0001	0.5 ^b^ ± 0.03	<0.0001
**IR control**	23.4 ^a^ ± 1.18	<0.0001	151.5 ^a^ ± 4.04	<0.0001	3.4 ^a^ ± 0.12	<0.0001
**OLP-10**	12.1 ^ab^ ± 0.46	<0.0001	83.0 ^ab^ ± 2.45	<0.0001	2.2 ^ab^ ± 0.10	<0.0001
**OLP-20**	6.3 ^b^ ± 0.45	0.4350	46.9 ^ab^ ± 2.33	0.0004	1.4 ^ab^ ± 0.07	<0.0001

Values are expressed as mean ± SEM of six animals in each group. ^a^ Significantly different from the values of sham rats at *p* ≤ 0.05. ^b^ Significantly different from the values of IR control rats at *p* ≤ 0.05.

**Table 2 ijms-25-10171-t002:** Effect of OLP treatment on oxidative stress biomarkers in pancreatic tissue of rats subjected to pancreatic ischemia/reperfusion (IR).

	MDA (nmol/mg Protein)	*p* Value	GPx (nmol/mg Protein)	*p* Value	MPO (ng/mg Protein)	*p* Value
**Sham**	0.3 ^b^ ± 0.02	<0.0001	3.7 ^b^ ± 0.20	<0.0001	0.9 ^b^ ± 0.06	<0.0001
**IR control**	1.9 ^a^ ± 0.04	<0.0001	0.6 ^a^ ± 0.03	<0.0001	7.5 ^a^ ± 0.31	<0.0001
**OLP-10**	1.1 ^ab^ ± 0.04	<0.0001	1.9 ^ab^ ± 0.09	<0.0001	5.2 ^ab^ ± 0.14	<0.0001
**OLP-20**	0.6 ^ab^ ± 0.03	<0.0001	3.4 ^b^ ± 0.13	0.1597	3.1 ^ab^ ± 0.16	<0.0001

Values are demonstrated as mean ± SEM of 6 animals/group. ^a^ values are different significantly from the sham rats at *p* ≤ 0.05. ^b^ values are different significantly from the IR control rats at *p* ≤ 0.05.

**Table 3 ijms-25-10171-t003:** Effect of OLP on inflammatory markers in pancreatic tissue of rats subjected to pancreatic ischemia/reperfusion (IR).

Group	TNF-*α* (pg/mg Protein)	*p* Value	IL-1*β* (pg/mg Protein)	*p* Value	NF*κB* (ng/mg Protein)	*p* Value
**Sham**	27.8 ^b^ ± 1.11	<0.0001	29.1 ^b^ ± 1.90	<0.0001	26.6 ^b^ ± 0.95	<0.0001
**IR control**	334.5 ^a^ ± 6.95	<0.0001	226.3 ^a^ ± 4.92	<0.0001	295.4 ^a^ ± 15.43	<0.0001
**OLP-10**	187.9 ^ab^ ± 2.10	<0.0001	143.2 ^ab^ ± 4.03	<0.0001	141.3 ^ab^ ± 1.84	<0.0001
**OLP-20**	58.5 ^ab^ ± 5.15	0.0005	60.5 ^ab^ ± 3.90	<0.0001	71.9 ^ab^ ± 4.76	0.0042

Values are expressed as mean ± SEM of six animals in each group. ^a^ Significantly different from the values of sham rats at *p* ≤ 0.05. ^b^ Significantly different from the values of IR control rats at *p* ≤ 0.05.

## Data Availability

The original contributions presented in the study are included in the article, further inquiries can be directed to the corresponding author.

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
