# Peer review of "Oleuropein Relieves Pancreatic Ischemia Reperfusion Injury in Rats by Suppressing Inflammation and Oxidative Stress through HMGB1/NF-κB Pathway"

_ijms, 2024, doi:10.3390/ijms251810171_

Round 1
Reviewer 1 Report
Comments and Suggestions for Authors
The article is interesting, however, the execution and presentation of this research contain several critical flaws that undermine its credibility and scientific value. Below are the major issues generated following review:
1. It is unclear how the author finds the development of liver tissue vacuolar degeneration within one hr of splenic artery occlusion. He didn't support his study with established protocols or discuss this part in the manuscripts.
2. Please include the exact individual p-value and mention it in figure legends or with bar graphs.
3. In experimental design, groups 2,3, and 4 should include ischemia reperfusion following, vehicle, 10mg/kg, and 2mg/kg to avoid confusion.
4. Conclusion should be included followed by discussion, instead of material method.
5. Lines 60-61 and 61-62 need some research article reference to support it.
6. The discussion section includes the methodology and results part, instead of discussing the study properly. Please remove that part and discuss the study with the supporting research study.
Author Response
- It is unclear how the author finds the development of liver tissue vacuolar degeneration within one hr of splenic artery occlusion. He didn't support his study with established protocols or discuss this part in the manuscripts.
In our study, splenic artery occlusion was done for 60 min, and the animals were sacrificed and blood and tissue samples were taken after 24 h.
As mentioned in the "Introduction", lines 47-51: “Pancreatic I/R injuries reported to increase blood and lavage white blood cell count, reactive oxygen species, and inflammatory cytokine release. As a result of acute pancreatitis, the development of the systemic inflammatory response syndrome, and multiple organ failure including the pancreas is a consequence of the release of inflammatory cytokines by activated leukocytes”.
This point is now made more clearly in the “Discussion”, lines 286–295:
Furthermore, in the rat model of severe acute pancreatitis, the spleen aggravates numerous organ damage and systemic inflammatory responses. Besides, rats with severe acute pancreatitis showed significantly increased ultrastructural damage to liver tissue, as evidenced by the following indicators: lysosome accumulation in Kuppfer cells, hepatocyte edema, mitochondrial swelling, nucleolar pyrknosis, cell apoptosis, and cell necrosis after 15 hours as determined by transmission electron microscopy (Zhou et al., 2019). These results confirm our observations that IR control revealed portal blood vessel congestion, as well as the presence of vacuolar degeneration in hepatocytes and low numbers of mononuclear inflammatory cells infiltrating the portal area.
Zhou R, Zhang J, Bu W, Zhang W, Duan B, Wang X, Yao L, Li Z, Li J. A new role for the spleen: Aggravation of the systemic inflammatory response in rats with severe acute pancreatitis. The American journal of pathology. 2019 Nov 1;189(11):2233-45.
- Please include the exact individual p-value and mention it in figure legends or with bar graphs.
Done.
- In experimental design, groups 2,3, and 4 should include ischemia reperfusion following, vehicle, 10mg/kg, and 2mg/kg to avoid confusion.
Done.
- Conclusion should be included followed by discussion, instead of material method.
Done.
- Lines 60-61 and 61-62 need some research article reference to support it.
Done.
- The discussion section includes the methodology and results part, instead of discussing the study properly. Please remove that part and discuss the study with the supporting research study.
The discussion section has been improved.
Reviewer 2 Report
Comments and Suggestions for Authors
The article “ Oleuropein relieves pancreatic ischemia-reperfusion injury in 2 rats by suppressing inflammation and oxidative stress through 3 HMGB1/NF-κB pathway “ written by Maged S. Abdel-Kader is a novel study focussing on the role of Oleuropein that can relieve pancreatic ischemia.
The study brings results on inflammatory panels, iNOS, and HMGH.
In my opinion, this paper requires some major concerns that need to be answered.
Major revision:
1. In some parts, relevant citations are not provided. For Eg: In line 111, current findings are written? What are these current studies? Or what studies are relevant?
2. Figure 6 does not have a proper figure legend. You must explain how you quantified iNOS-positive cells.
3. Even if you discuss major findings and their relevance in the discussion, it is necessary to write a few words about that in the text before going to figure out and explain the results that you got. It is not clear why you specifically focussed on HMGB1, iNOS, and so on.
4. Have you tried any other antioxidant levels like GPx? If yes, please include it as a supplementary figure.
5. In Materials and Methods: Some parts is missing. For Eg: In Serum analyses, line 343, its written according to the manufacturer. Who is the manufacturer? Complete the manufacturer details in other missing parts as well.
English
1. Minor mistakes were found in the manuscript. Small corrections in terms of grammar and language needs to be done.
For Eg: Line 113-HMGB1mRNA and Line 114 HMGB1expression are written as a single word.
Comments on the Quality of English LanguageEnglish usage and sentence formation needs to be improved.
Author Response
- In some parts, relevant citations are not provided. For Eg: In line 111, current findings are written? What are these current studies? Or what studies are relevant?
Done.
- Figure 6 does not have a proper figure legend. You must explain how you quantified iNOS-positive cells.
Done.
- Even if you discuss major findings and their relevance in the discussion, it is necessary to write a few words about that in the text before going to figure out and explain the results that you got. It is not clear why you specifically focussed on HMGB1, iNOS, and so on.
The discussion section has been improved.
- Have you tried any other antioxidant levels like GPx? If yes, please include it as a supplementary figure.
We measured oxidative stress markers (MDA, MPO and GPX) in pancreatic tissue. And we have supported our findings with previous records of the antioxidant effect of OLP in other research work within the manuscript.
- In Materials and Methods: Some parts is missing. For Eg: In Serum analyses, line 343, its written according to the manufacturer. Who is the manufacturer? Complete the manufacturer details in other missing parts as well.
Done.
English
- Minor mistakes were found in the manuscript. Small corrections in terms of grammar and language needs to be done.
For Eg: Line 113-HMGB1mRNA and Line 114 HMGB1expression are written as a single word.
Done.
Comments on the Quality of English Language
English usage and sentence formation needs to be improved.
Done.
Round 2
Reviewer 1 Report
Comments and Suggestions for Authors
The authors have satisfactorily responded to my concerns.
Reviewer 2 Report
Comments and Suggestions for Authors
Dear Authors,
The article "Oleuropein relieves pancreatic ischemia-reperfusion injury in 2 rats by suppressing inflammation and oxidative stress through 3 HMGB1/NF-κB pathway" is much better than the previous version.